# DS-VLM: Diffusion Supervision Vision Language Model

Zhen Sun[1]  Yunhang Shen[2]  Jie Li[3]  Xing Sun[2]  Pingyang Dai[1]  Liujuan Cao[1]  Rongrong Ji[1 4]

## Abstract

Vision-Language Models (VLMs) face two critical limitations in visual representation learning: degraded supervision due to information loss during gradient propagation, and the inherent semantic sparsity of textual supervision compared to visual data. We propose the Diffusion Supervision Vision-Language Model (DS-VLM), a plug-and-play framework that introduces diffusion-based direct supervision for vision-language alignment. By reconstructing input images through a diffusion model conditioned on outputs of the visual encoder and the connector, our method establishes a short-path gradient propagation channel from pixel space to visual features. This approach simultaneously preserves high-level semantic alignment through conventional text supervision while enhancing visual feature quality via pixel-level reconstruction constraints. Extensive experiments conducted across various visual encoders and LLMs of different scales demonstrate the effectiveness of our approach.

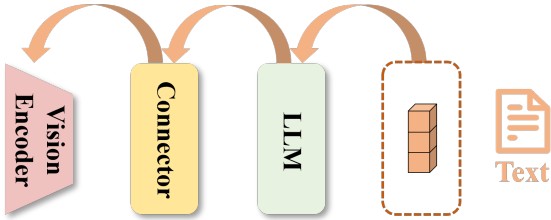

(a) The text knowledge propagation chain of current methods.

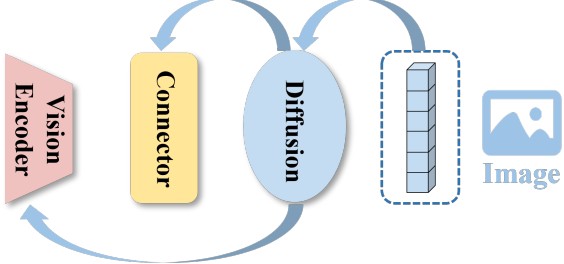

(b) The image knowledge propagation chain of the proposed DS-VLM.

*Figure 1.* The figure illustrates the knowledge propagation process of current methods and our proposed DS-VLM. In comparison, the DS-VLM has a shorter propagation chain and leverages the rich semantic details contained in the image.

## 1. Introduction

Large language models (LLMs), exemplified by ChatGPT, have achieved remarkable progress in text understanding and generation. Meanwhile, multimodal large language models (MLLMs) have further extended the capabilities of LLMs by incorporating image understanding, evolving into models capable of integrating visual and textual modalities. This development has made MLLMs a new focal point for research and discussion.

In general terms, the architecture of mainstream MLLMs (Li

[1]Key Laboratory of Multimedia Trusted Perception and Efficient Computing, Ministry of Education of China, Xiamen University, 361005, P.R. China. [2]Tencent YouTu Lab [3]School of Informatics, Xiamen University, Xiamen, China. [4]Institute of Artificial Intelligence,Xiamen University,Xiamen,China. Correspondence to: Liujuan Cao <caoliujuan@xmu.edu.cn>.

*Proceedings of the 42$^{nd}$ International Conference on Machine Learning*, Vancouver, Canada. PMLR 267, 2025. Copyright 2025 by the author(s).

et al., 2023b) can be delineated into three components: the pre-trained vision encoder (Radford et al., 2021; Sun et al., 2023), the pre-trained LLM (Zhang et al., 2023c; Touvron et al., 2023; Chiang et al., 2023), and the connector (Alayrac et al., 2022; Li et al., 2023b; Liu et al., 2024b;a) trained from scratch to bridge the vision and language models.

The most critical factor affecting the performance of models in understanding multi-modal content lies in the effectiveness of the vision encoder and connector in processing image features. Therefore, most current methods focus on improving the structure of either the Vision Encoder or the Connector components. For example, Cumo (Li et al., 2024) applies the Mixture of Experts (MoE) mechanism to MLLMs by integrating Top-K sparse-gating MoE blocks into the multilayer perceptron (MLP) connector. BLIP-2 (Li et al., 2023b) introduces Q-Former, which leverages a set of learnable query vectors to filter and refine raw visual features, extracting visual information relevant to language tasks. Similarly, MG-LLava (Zhao et al., 2024b) integrates object-level features by incorporating bounding boxes iden-

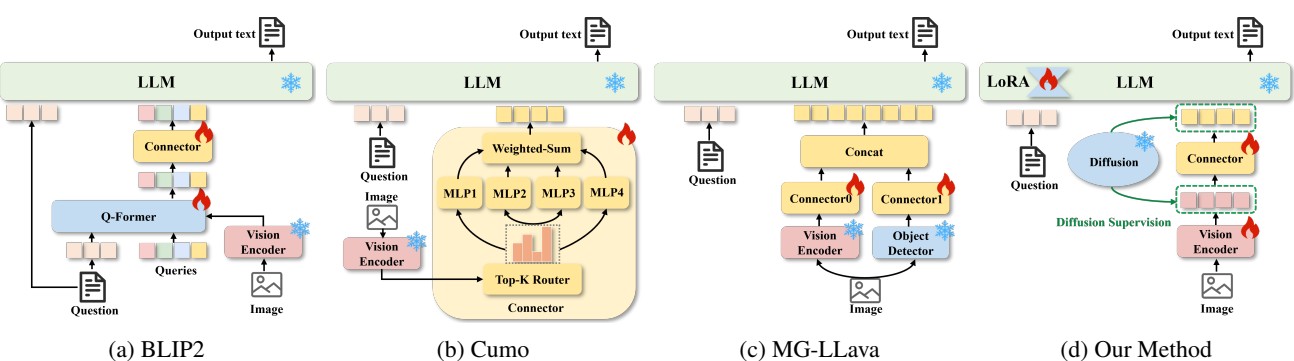

*Figure 2.* The figure illustrates the existing mainstream methods and our proposed method. (a), (b), and (c) represent the existing mainstream methods, while (d) shows our proposed DS-VLM. It supervises the Vision Encoder and Connector by introducing a diffusion model during the training phase, which shortens the knowledge propagation chain and does not require the addition of extra components during the inference phase.

tified by an offline detector, aiming to enhance the model's object recognition capability, as illustrated in Figure 2c. In these methods, the optimization of the visual encoder and connector components typically relies on backpropagation through textual information during training, as illustrated in Figure 1a. Specifically, during training, the gradients computed from the next-token prediction loss between the predicted answers generated by the LLM and the target text are backpropagated through the LLM and the connector to the visual encoder. However, this approach suffers from an excessively long knowledge propagation chain, particularly when the path traverses the LLM with its massive number of parameters, leading to significant optimization knowledge loss. This hinders effective parameter updates and optimization of these two components. Moreover, as the saying goes, "a picture is worth a thousand words," the information contained in text is far less rich than that in images, especially in terms of semantic details related to image content, which text struggles to fully capture. Therefore, exploring methods to directly leverage image information for optimizing the visual encoder and connector components while shortening the knowledge propagation chain is a highly significant yet challenging research direction.

Based on this, we propose a plug-and-play method, Diffusion Supervision Vision Language Model (DS-VLM), which leverages a pretrained diffusion model to supervise the vision encoder and the connector during the training phase, as shown in Figure 2d. The knowledge backpropagation path in this method allows image information to propagate directly from the diffusion model to the vision encoder and connector components, bypassing the LLM, thereby significantly shortening the propagation path. Specifically, we treat the features output by the vision encoder as image features and the features output by the connector as text feature since the connector maps image feature the space of text embeddings. Inspired by (Yao et al., 2024), we recognize

the importance of the semantic information contained in the intermediate features of the vision encoder for enhancing the performance of vision-language models (VLMs). Therefore, we apply supervision to the intermediate and final layer features of the vision encoder as well as to the output features of the connector. To adapt the diffusion model to the distinct supervised features of the two modalities, we designed a Multi-Adapter Diffusion model based on the Stable Diffusion framework. This model can generate images conditioned on the supervised features. Finally, we construct a reconstruction loss to measure the differences between the generated image and the original image. Through optimization, this approach ensures that the extracted image features comprehensively capture the semantic, structural, and other critical information of the original image. During the inference phase, the Diffusion model is unnecessary, thereby avoiding the increase in number of model parameters and computational overhead compared to (Li et al., 2024; 2023b; Zhao et al., 2024b). Extensive experiments demonstrate the effectiveness of the proposed design, showing significant improvements over the baseline across multiple datasets and frameworks.

The main contributions are summarized as follows:

- We propose the DS-VLM method, which introduces a diffusion model to leverage the rich semantic information of images for optimizing the visual encoder and connector through a shorter knowledge propagation chain. Our method brings richer supervision signal and higher optimization efficiency, without extra cost for inference.

- We design Multi-Adapter Diffusion to receive supervised features from different modalities and reconstruct the original image. Serving as a medium for knowledge propagation, it transmits the semantic information contained in the image to the Vision Encoder and Con-

nector components.

- Extensive experiments conducted on various visual encoders and LLMs of different scales validate the effectiveness of our method, demonstrating that the proposed DS-VLM not only brings substantial benefits but also provides valuable insights and approaches for the existing VLM research field.

## 2. Related Work

### 2.1. Vision Language Models

Vision-Language Models (VLMs) are designed to simultaneously process visual and linguistic information. With the growing demand for multimodal tasks, VLMs have been widely applied to tasks such as image captioning (Chen et al., 2022b; Dai et al., 2023; Wang et al., 2024), visual question answering (VQA) (Xu et al., 2023; Yu et al., 2022), and image-text retrieval (Ge et al., 2024; Yang et al., 2024b) . They represent prominent architectures in the multimodal domain. Researchers have proposed numerous architectures (Li et al., 2023a; Zhu et al., 2024; Chen et al., 2023) for integrating visual features into advanced LLM inference pipelines. Llama-Adapter (Zhang et al., 2023b) proposes to generate language answer with taking the image input as condition. Flamingo (Alayrac et al., 2022) and LLaVA (Liu et al., 2024b) blend visual tokens with text as inputs to LLM, differing in that Flamingo employs gating mechanisms to inject encoded visual features into LLMs, while LLaVA directly concatenates visual and textual features at input. In addition, BlIP2 (Li et al., 2023b) introduces Q-Former, which extracts vision information relevant to language tasks through a set of learnable query vectors. Cumo (Li et al., 2024) applies MOE to MLLM, integrating Top-K sparse gated MOE blocks within a multi-layer perceptron (MLP) connector. MG-LLava (Zhao et al., 2024b) introduces object-level features by incorporating bounding boxes detected by an offline detector. The optimization process of these methods involves propagating the textual information through the LLM and Connector, eventually reaching the Vision Encoder, resulting in a extended optimization chain. To address this, we propose an optimization method with a shorter chain, DS-VLM, which uses Diffusion as an intermediary to directly propagate image knowledge to the Vision Encoder.

### 2.2. Diffusion Models

**Text-to-Image Diffusion Models** Recently, diffusion models (Sohl-Dickstein et al., 2015; Song et al., 2020a;b; Dhariwal & Nichol, 2021) have emerged as the new state-of-the-art model for text-to-image generation. As a pioneer, GLIDE (Nichol et al., 2021) uses a cascaded diffusion architecture with a 3.5B text-conditional diffusion model at 64×64 resolution and a 1.5B text-conditional upsampling diffusion model at 256×256 resolution. DALL-E 2 (Ramesh et al., 2022) employs a diffusion model conditioned image embedding, and a prior model was trained to generate image embedding by giving a text prompt. DALL-E 2 not only supports text prompt for image generation but also image prompt. To enhance the text understanding, Imagen (Saharia et al., 2022) adopts the T5 model (Raffel et al., 2020), a large transformer language model pretrained on text-only data, as the text encoder of diffusion model. Re-Imagen (Chen et al., 2022a) uses retrieved information to improve the fidelity of generated images for rare or unseen entities. Stable Diffusion (SD) (Rombach et al., 2022) is built on the latent diffusion model (Rombach et al., 2022) , which operates on the latent space instead of pixel space, enabling SD to generate high-resolution images with only a diffusion model. In this paper, we design Multi-Adapter Diffusion based on the SD model, which can receive supervisory features from two modalities and generate images conditioned on them.

**Adapters for Text-to-Image Models** With the popularity of recent text-to-image models, adapters have also been used to provide additional control for the generation of text-to-image models. ControlNet (Zhang et al., 2023a) first proves that an adapter could be trained with a pretrained text-to-image diffusion model to learn task-specific input conditions, e.g., canny edge. Almost concurrently, T2I-adapter (Mou et al., 2024) employs a simple and lightweight adapter to achieve fine-grained control in the color and structure of the generated images. To reduce the fine-tuning cost, Uni-ControlNet (Zhao et al., 2024a) presents a multi-scale condition injection strategy to learn an adapter for various local controls. In the updated version of T2I-adapter, a style adapter is designed to control the style of generated images using a reference image by appending image features extracted from the CLIP image encoder to text features. The global control adapter of Uni-ControlNet also projects the image embedding from CLIP image encoder into condition embeddings by a small network and concatenates them with the original text embeddings, and it is used to guide the generation with the style and content of reference image. In this study, we introduce Multi-Adapter Diffusion, which incorporates a separated cross-attention mechanism to receive supervisory features from two modalities, enabling more effective image reconstruction.

## 3. Method

### 3.1. Preliminary

As one of the most extensively adopted multi-modal LLM architectures, LLaVA consists of a vision encoder $f_V$, an MLP projector $f_p$, and a large language model $f_L$. Given a visual input $V$ and a textual input $T$, LLaVA computes the vision and language embeddings as per Eq. (1), where $f_T$

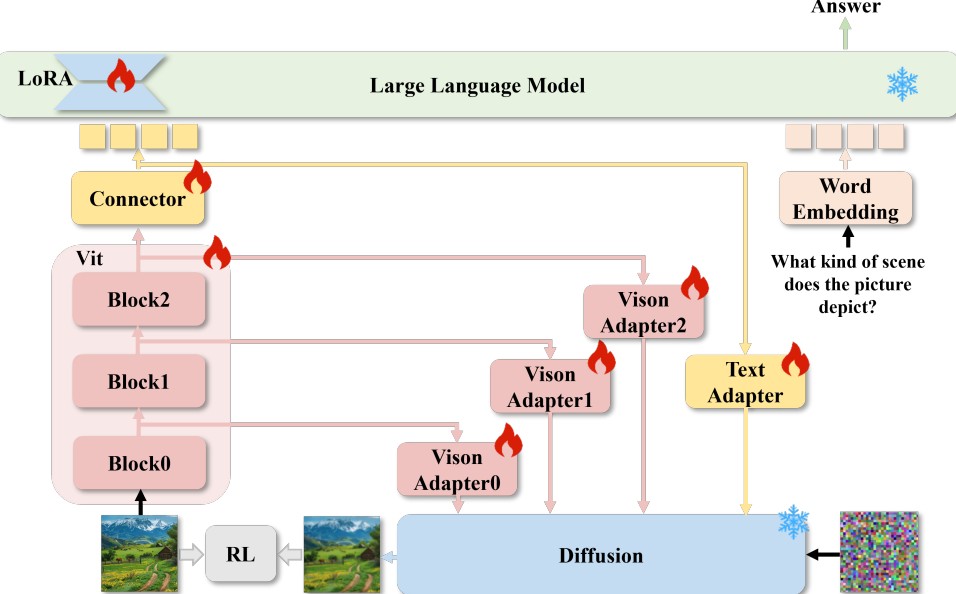

*Figure 3.* The overall framework of the proposed DS-VLM. It uses a diffusion model to supervise the output features of the vision encoder and connector. The high-level, mid-level, and low-level features of the vision encoder are treated as image modality features, while the output features of the connector are treated as text modality features. These features are input into their respective adapters and processed as conditional features to assist the diffusion model in reconstructing the original image. RL represents the reconstruction loss.

represents the tokenizer of $f_L$. The resulting embeddings, $\mathbf{E_T}$ and $\mathbf{E_V}$, are then concatenated into a single token sequence, serving as the input to the LLM. LLaVA utilizes Eq. (2) to calculate the probability of the target answer $X_A$, where $\theta$ represents the trainable parameters and $L$ is the length of $X_A$. The model is trained on visual instruction tuning data to maximize $p(\mathbf{X_A}|V, T)$.

$$\mathbf{E}_T = f_T(T), \mathbf{E}_V = f_p(f_V(V)) \tag{1}$$

$$p(X_A|V, T) = \prod_{i=1}^{L} p_\theta(X_A^{[i]}|\text{Concat}(\mathbf{E}_V, \mathbf{E}_T^{[1:i-1]}), X_A^{[i-1]}) \tag{2}$$

Despite the encouraging results, most current methods (Li et al., 2023b; 2024; Zhao et al., 2024b) focus on optimizing the structure of the visual encoder and connector components or adding additional components to enhance the performance of extracted features. The optimization of these two components typically relies on backpropagation through textual information during training. Specifically, gradients propagate through the LLM and Connector to the vision encoder. However, this results in an excessively long knowledge propagation chain, particularly when the path

traverses an LLM with a large number of parameters, leading to significant optimization knowledge loss. Moreover, the information contained in text is far less rich compared to images. To address these issues, we propose DS-VLM, as illustrated in Figure 3, which introduces Diffusion to supervise the vision encoder and connector components. The diffusion model reconstructs the original image based on the output features of these two components and optimizes them through reconstruction loss. Image knowledge is directly propagated to these two components via Diffusion, significantly shortening the backpropagation path. At the same time, it leverages the richer semantic content that images can provide compared to text, thereby greatly improving the quality of the extracted features and enhancing the overall performance of the model.

### 3.2. Diffusion Supervision

**Supervision Features** First, it is essential to determine which features should be input into the diffusion model as conditions for reconstructing noise. Specifically, we classify the supervision features into two categories: image supervision features and text supervision features. For image supervision features, inspired by (Yao et al., 2024), which highlights the importance of intermediate layer features in the vision encoder for improving the understanding capabilities of vision-language models (VLMs), and to more comprehensively supervise the vision encoder features, we

extract low, mid and high layer features from the vision encoder as supervision features for image. The connector maps image features into the text feature domain, and therefore, its output features naturally serve as the supervision features for text.

**Multi-Adapter Diffusion.** Diffusion models are a general framework for generative modeling. In recent years, they have received widespread attention due to their outstanding performance in the field of image generation. They consist of two processes: a diffusion process (also known as the forward process), which gradually adds Gaussian noise to the data using a fixed Markov chain of $T$ steps, and a denoising process that generates samples from Gaussian noise with a learnable model. Diffusion models can also be conditioned on other inputs, such as text in the case of text-to-image diffusion models. Typically, the training objective of a diffusion model, denoted as $\epsilon_\theta$, which predicts noise, is defined as a simplified variant of the variational bound:

$$L_{\text{simple}} = \mathbb{E}_{x_0, \epsilon \sim \mathcal{N}(0, \mathbf{I}), c, t} \|\epsilon - \epsilon_\theta(x_t, c, t)\|^2 \quad (3)$$

where $x_0$ represents the real data with an additional condition $c$, $t \in [0, T]$ denotes the time step of diffusion process, $x_t = \alpha_t x_0 + \sigma_t \epsilon$ is the noisy data at $t$ step, and $\alpha_t$, $\sigma_t$ are predeffned functions of $t$ that determine the diffusion process. In this study, we use the open-source SD model as the base model to construct our Multi-Adapter Diffusion. SD is a latent diffusion model conditioned on text features extracted from a frozen CLIP text encoder. The architecture of the diffusion model is based on a UNet (Ronneberger et al., 2015) with attention layers. In the original SD model, the text features from the CLIP text encoder are plugged into the UNet model by feeding into the cross-attention layers. Given the query features $\mathbf{Z}$ and the text features $\mathbf{c}_t$, the output of cross-attention $\mathbf{Z}'$ can be defined by the following equation:

$$\mathbf{Z}' = \text{Attention}(\mathbf{Q}, \mathbf{K}, \mathbf{V}) = \text{Softmax}(\frac{\mathbf{Q}\mathbf{K}^T}{\sqrt{d}})\mathbf{V} \quad (4)$$

where $\mathbf{Q} = \mathbf{Z}\mathbf{W}_q$, $\mathbf{K} = \mathbf{c}_t\mathbf{W}_k$, $\mathbf{V} = \mathbf{c}_t\mathbf{W}_v$ are the query, key, and values matrices of the attention operation respectively, and $\mathbf{W}_q$, $\mathbf{W}_k$, $\mathbf{W}_v$ are the weight matrices of the trainable linear projection layers. Inspired by SD model, We propose Multi-Adapter Diffusion, as shown in Figure 4, which utilizes multiple adapters to receive and preprocess features from both image and text modalities. The processed features are then fed into the U-Net network through an MOE Cross Attention mechanism to assist the Diffusion model in reconstructing the original image. The MOE Cross Attention mechanism will be detailed in Section 3.3.

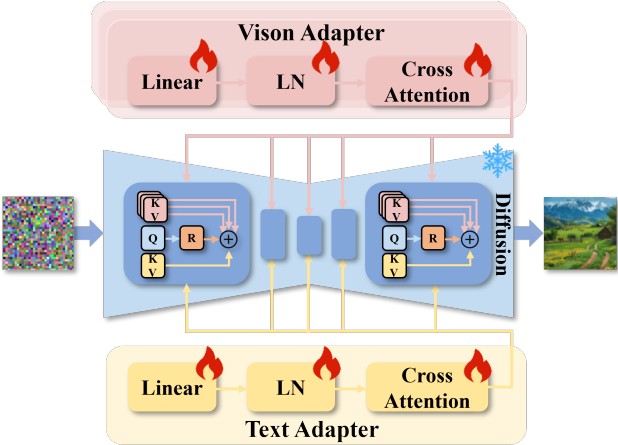

*Figure 4.* Illustration of Multi-Adapter Diffusion, which employs multiple adapters to receive different supervised features and integrates these features into the Diffusion model through the MOE Cross Attention mechanism.

### 3.3. MOE Cross Attention

A straightforward method to incorporate image features is to concatenate the image features from various layers of the vision encoder with text features and then feed them into a cross-attention layer. However, we found this approach to be suboptimal. Instead, we propose an MOE (Mixture-of-Experts) cross attention mechanism, as shown in Figure 4, where the cross attention layers for text features and image features from different layers are separated. The outputs of all these features are aggregated using an MOE routing mechanism. Specifically, for each cross-attention layer in the original U-Net model, we add three new cross-attention layers to integrate low, mid and high level image features. Given the image feature $c_i$, where $i$ is the layer index. the output of new cross-attention $Z_i''$ is compute as follows:

$$Z_i'' = \text{Attention}(\mathbf{Q}, \mathbf{K}_i', \mathbf{V}_i') = \text{Softmax}(\frac{\mathbf{Q}(\mathbf{K}_i')^T}{\sqrt{d}})\mathbf{V}_i' \quad (5)$$

where, $\mathbf{Q} = \mathbf{Z}\mathbf{W}_q$, $\mathbf{K}_i' = c_i\mathbf{W}_{ki}'$ and $\mathbf{V}_i' = c_i\mathbf{W}_{vi}'$ are the query, key, and values matrices from the i-th layer image features. $\mathbf{W}_{ki}'$ and $\mathbf{W}_{vi}'$ are the corresponding weight matrices. It should be noted that we use the same query for image cross-attention as for text cross-attention. In order to speed up the convergence, $\mathbf{W}_{ki}'$ and $\mathbf{W}_{vi}'$ are initialized from $\mathbf{W}_k$ and $\mathbf{W}_v$. Then we employ the MOE routing algorithm to compute the weights for the output feature of each cross-attention, followed by a weighted summation to obtain the new feature representation. Accordingly, the final formulation of the MOE cross attention is defined as follows:

$$\mathbf{P} = \text{Softmax}(\text{Pooling}(\mathbf{Q} \cdot \mathbf{W}_r)) \quad (6)$$

*Table 1.* Performance comparison between various baselines and DS-VLM. The best results are **bold**. LM, VE, PT and IT denote Language Model, Vision Encoder, pre-training data and instruction fine-tuning data, respectively.

| Method | LM | VE | PT + IT | MMB | MMS | MMMU | MV | OCRB | AI2D | HB | LB | SQA | MME |
|--------|-----|-----|---------|------|------|------|------|------|------|------|------|------|------|
| **Performance comparison against the baseline** | | | | | | | | | | | | | |
| LLaVA-1.5 | Vicuna-7B | CLIP-L | 0.5 M + 0.6M | 59.1 | 33.1 | 35.7 | 25.6 | 31.8 | 55.5 | 27.6 | 61.8 | 69.2 | **1808** |
| DS-VLM | Vicuna-7B | CLIP-L | 0.5 M + 0.6M | **61.5** | **34.9** | **37.8** | **27.1** | **32.4** | **56.7** | **28.9** | **64.3** | **69.7** | 1783 |
| LLaVA-1.5 | Vicuna-13B | CLIP-L | 0.5 M + 0.6M | 64.0 | 34.3 | 37.0 | 27.7 | **33.7** | 61.1 | 24.5 | 66.1 | 72.6 | 1781 |
| DS-VLM | Vicuna-13B | CLIP-L | 0.5 M + 0.6M | **65.5** | **35.8** | **38.9** | **28.6** | 33.5 | **62.2** | **25.7** | **67.9** | **73.7** | 1836 |
| **Expanding to larger training datasets** | | | | | | | | | | | | | |
| LLaVA-1.5 | Vicuna-7B | CLIP-L | 1.2 M + 1.5M | 62.8 | 39.0 | 35.2 | 32.6 | **37.3** | 69.8 | **25.4** | 60.7 | 70.5 | 1810 |
| DS-VLM | Vicuna-7B | CLIP-L | 1.2 M + 1.5M | **63.6** | **39.3** | **37.5** | **33.2** | 36.8 | **70.1** | 25.2 | **61.4** | **71.3** | 1857 |
| **Expanding to robust visual encoder** | | | | | | | | | | | | | |
| LLaVA-1.5 | Vicuna-7B | SigLIP-SO | 0.5 M + 0.6M | 62.8 | 34.9 | **38.6** | 27.0 | 36.3 | 59.3 | **28.1** | 66.8 | 70.6 | 1764 |
| DS-VLM | Vicuna-7B | SigLIP-SO | 0.5 M + 0.6M | **63.2** | **36.8** | 38.4 | **27.6** | **37.5** | **60.0** | 27.8 | **67.7** | **71.1** | **1814** |
| **Expanding to other LLMs** | | | | | | | | | | | | | |
| LLaVA-1.5 | Llama3-8B | CLIP-L | 0.5 M + 0.6M | **66.7** | 38.5 | 40.7 | 26.7 | **33.4** | 61.8 | 27.4 | 64.3 | **74.8** | 1789 |
| DS-VLM | Llama3-8B | CLIP-L | 0.5 M + 0.6M | 66.3 | **40.4** | **41.5** | **28.3** | 32.5 | **62.0** | **28.6** | **65.1** | 74.3 | **1825** |
| LLaVA-1.5 | Qwen2-7B | CLIP-L | 0.5 M + 0.6M | **70.9** | 42.1 | 43.6 | 32.2 | **33.6** | **65.3** | 28.3 | 65.9 | 74.2 | 1849 |
| DS-VLM | Qwen2-7B | CLIP-L | 0.5 M + 0.6M | 70.4 | **43.7** | **44.8** | **33.0** | 33.2 | 64.6 | **29.5** | **66.4** | **75.0** | **1953** |

$$\mathbf{Z}^{new} = \mathbf{P}_0\mathbf{Z}' + \sum_{i=1}^{3} \mathbf{P}_i\mathbf{Z}_i'' \qquad (7)$$

where $\mathbf{W}_r \in \mathbf{R}^{d \times 4}$ is a learnable parameter and $d$ is the hidden dimension of the feature. $\mathbf{P}$ is a vector of size 4 and $\mathbf{P}_0$ represents the weight of the output features from the textual Cross Attention, while $\mathbf{P}_1$, $\mathbf{P}_2$, and $\mathbf{P}_3$ represent the weights of the output features from the low, mid and high visual Cross Attention layers, respectively.

### 3.4. Reconstruction Loss

As diffusion models restore images from noise, selecting an effective loss function to measure the difference between the original and generated images becomes critical. Here, we attempt to construct reconstruction losses at three levels of granularity: low-level, mid-level and high-level. Specifically, we employ mean absolute error (MAE) for the pixel level, structural similarity index (SSIM) for the structural level, and perceptual loss (PL) for the semantic level. By comparing the effects of these three reconstruction losses, we aim to comprehensively evaluate the model's reconstruction performance and identify the most suitable loss function to optimize the model. Unless otherwise specified, all the experiments and discussions in this paper are conducted using the perceptual loss.

## 4. Experiment

In this section, we first present the detailed experimental setup of our study. We then enumerate the improvements brought by our proposed DS-VLM over the baseline across multiple evaluation metrics, and compare our method with several state-of-the-art (SoTA) approaches under various configurations. We also conduct ablation studies and ana-

lyze the results. Finally, we visualize the denoising process of the diffusion model to demonstrate the effectiveness of the proposed DS-VLM.

### 4.1. Experimental Settings

**Implementation Details** We implement the proposed improvement strategy on top of LLaVA-1.5 (Liu et al., 2024a), whose general applicability in the VLM field facilitates the validation of our method's versatility. Specifically, we maintain consistency with LLaVA-1.5 by employing CLIP-ViTL/14-336px as the visual encoder. To further validate the generalizability of our proposed method, we also incorporate SigLIP-SO400m-patch14-384 (Zhai et al., 2023), another leading choice, for comparative analysis. In terms of LLM, we compare our method against the baseline using Vicuna-7/13B and extend our approach to Llama3-8B (Meta, 2024) and Qwen2-7B (Yang et al., 2024a), thereby demonstrating the versatility of our method. For training configurations, we adhere strictly to the settings outlined in the original LLaVA-1.5 paper to ensure fairness, with learning rates of 1e-3 and 2e-5 for pre-training and instruction fine-tuning phases, respectively, and maintaining batch sizes of 256 and 128. During the LoRA fine-tuning process, the rank of all linear layers is uniformly set to 8. We select the 8th, 16th, and 24th layers of the Vision encoder as the feature representatives of the low, mid and high layers, respectively. The number of iterations for the diffusion model is 50. The training process for DS-VLM utilizes the PyTorch framework.

**Datasets** Focusing on proposing a novel optimization method for the VLM framework, we do not incorporate any additional data beyond the LLaVA-1.5 open-source dataset (Liu et al., 2024a), which includes 558K image captions for pre-training and 665K conversations for in-

*Table 2.* Comparison with SoTA methods. The best results are **bold** and the second-best results are underlined.

| Method | LM | VE | PT + IT | MMB | MMS | MMMU | MV | OCRB | AI2D | HB | LB | SQA | MME |
|---|---|---|---|---|---|---|---|---|---|---|---|---|---|
| MiniGPT4 | Vicuna-7B | EVA-G | 5 M + 3.5K | 20.8 | 16.3 | 23.6 | 20.4 | 17.2 | 28.4 | 31.9 | 45.1 | 39.6 | 1047 |
| Qwen-VL | Qwen-7B | ViT-G/16 | 1.4 B + 50M | 32.9 | 32.5 | 29.6 | 15.5 | 12.7 | 57.7 | 29.9 | 12.9 | 61.1 | 483 |
| VisualGLM | ChatGLM-6B | EVA-CLIP | 330 M | 35.7 | 25.9 | 29.9 | 21.9 | 17.0 | 41.2 | 25.0 | 37.3 | 56.1 | 738 |
| PandaGPT | Vicuna-13B | IB-H | 160 K | 34.5 | 25.6 | 32.9 | 25.0 | 26.9 | 48.3 | 21.6 | 57.2 | 61.8 | 1076 |
| mPLUG-Owl2 | Llama 2-7B | CLIP-L | 348 M + 1.2M | 60.8 | 34.8 | 34.7 | 25.4 | 25.5 | 55.7 | 29.4 | 59.9 | 69.5 | 1786 |
| Emu2-chat | Llama-33B | EVA-CLIP | - | 52.8 | 40.7 | 35.0 | 30.7 | **43.6** | 49.7 | 29.5 | 56.4 | 68.2 | 1678 |
| Yi-VL | Yi-6B | CLIP-L | 100 M + 26M | 64.2 | 33.7 | 40.3 | 29.7 | 29.0 | 59.8 | **36.0** | 51.9 | 72.6 | 1915 |
| ShareGPT-4V | Vicuna-7B | CLIP-L | 1.2 M + 0.7M | 61.6 | 35.7 | 37.2 | 26.5 | 37.1 | 58.0 | 28.6 | 66.9 | 69.5 | 1914 |
| LLaVA-1.5 | Qwen2-7B | CLIP-L | 0.5 M + 0.6M | **70.9** | 42.1 | 43.6 | 32.2 | 33.6 | 65.3 | 28.3 | 65.9 | 74.2 | 1849 |
| DS-VLM | Vicuna-7B | SigLIP-SO | 0.5 M + 0.6M | 63.2 | 36.8 | 38.4 | 27.6 | 37.5 | 60.0 | 27.8 | **67.7** | 71.1 | 1814 |
| DS-VLM | Vicuna-7B | CLIP-L | 1.2 M + 1.5M | 63.6 | 39.3 | 37.5 | **33.2** | 36.8 | **70.1** | 25.2 | 61.4 | 71.3 | 1857 |
| DS-VLM | Llama3-8B | CLIP-L | 0.5 M + 0.6M | 66.3 | 40.4 | 41.5 | 28.3 | 32.5 | 62.0 | 28.6 | 65.1 | 74.3 | 1825 |
| DS-VLM | Qwen2-7B | CLIP-L | 0.5 M + 0.6M | 70.4 | **43.7** | **44.8** | 33.0 | 33.2 | 64.6 | 29.5 | 66.4 | **75.0** | **1953** |

struction tuning. We also apply our proposed method to the Mini-Gemini dataset (Team), which consists of 1.2M + 1.5M data, to further highlight the superiority of our approach. For evaluation, we conduct extensive experiments on widely-adopted VLM benchmarks, providing robust and comprehensive performance validation for the proposed DS-VLM. The evaluation datasets include: MMBench (MMB) (Liu et al., 2025), MMS (MM-Star) (Chen et al., 2024), MMMU (Yue et al., 2024), MV (MathVista) (Lu et al., 2023), OCRB (OCRBench) (Liu et al., 2023), AI2D (Hiippala et al., 2021), HB (HallusionBench) (Guan et al., 2024), LB (LLaVABench) (Liu et al., 2024b), SQA (ScienceQA) (Saikh et al., 2022), and MME (Fu et al., 2024).

### 4.2. Improvement Over the Baseline

In table 1, we present the performance improvements of the proposed method across various configurations compared to the baseline. According to the experimental results, we can draw several phenomenons:

- The proposed DS-VLM demonstrates substantial improvements over the baseline. Compared with the original LLaVA-1.5, DS-VLM achieve much better performance. As shown in the first four rows. our method leads on the majority of evaluation datasets. It is noteworthy that DS-VLM demonstrates an average improvement of 1.4% over the original LLaVA1.5 across ten datasets when using Vicuna-7B, highlighting the method's significant value. When compared with the baseline LLaVA-1.5 Vicuna-7B model, we enhance performance metrics by +2.4% on MMBench, +2.1% on MMMU, and +2.5% on LLaVABenchs, respectively. Specifically, MMMU involves complex visual reasoning across multidisciplinary domains such as medicine and engineering, which requires precise interpretation of structural visual content and domain-specific symbols. DS-VLM excels in such scenarios by leveraging pixel-level reconstruction to enhance vi-

sual feature representations. MMBench emphasizes comprehensive multimodal understanding, including layout reasoning and structured chart interpretation. Our method's multi-level supervision effectively improves the model's sensitivity to spatial and structural details. Furthermore, LLaVABench emphasizes not only multimodal alignment but also the model's ability to perform knowledge grounding in complex real-world scenarios. DS-VLM addresses these challenges by introducing pixel-level reconstruction supervision, which enhances the semantic richness and fidelity of visual features. For LLaVA-1.5 with Vicuna-13B, we also achieve an average performance improvement of 1%. Specifically, we see a +1.5% gain on MMStar, a +1.9% gain on MMMU, and a +1.2% gain on HallusionBench. The improvement on MMStar, which emphasizes fine-grained multimodal alignment and reasoning under instruction-following settings, demonstrates DS-VLM's ability to preserve semantic coherence across modalities through enhanced visual feature quality. Meanwhile, HallusionBench is designed to assess a model's robustness against hallucinated or misleading visual-linguistic content. The stronger visual representations induced by diffusion supervision help DS-VLM better anchor its predictions to grounded visual evidence, effectively reducing semantic drift and hallucination. These impressive results further validate the contribution of the proposed DS-VLM architecture to visual feature optimization, highlighting the favorable impact of our method.

- The proposed Diffusion-based supervision method can be widely applied as a modular plugin within mainstream VLM frameworks. As shown in the rest part of Table 1. we further validate the versatility of our proposed method under various settings. We replace CLIP with SigLIP and substitute Vicuna with Llama3 and Qwen2 on top of the original LLaVA-1.5 framework. We compare these settings with our method as the baseline. The results in Table 1 confirm that

*Table 3.* Ablation study results on different types of supervision features, where TS represents text modality supervision and IS represents image modality supervision.

| No. | Components | | | | MMB | MMS | MMMU | MV | OCRB | AI2D | HB | LB | SQA | MME |
| | TS | IS (High) | IS (Mid) | IS (Low) | | | | | | | | | | |
|---|---|---|---|---|---|---|---|---|---|---|---|---|---|---|
| 0 | | | | | 64.0 | 34.3 | 37.0 | 27.7 | 33.7 | 61.1 | 24.5 | 66.1 | 72.6 | 1781 |
| 1 | ✓ | | | | 64.3 | 34.9 | 37.4 | 28.0 | 32.8 | 61.4 | 24.8 | 66.5 | 72.9 | 1798 |
| 2 | | ✓ | | | 64.8 | 35.2 | 38.0 | 27.9 | 33.1 | 61.6 | 24.6 | 66.9 | 72.8 | 1804 |
| 3 | | ✓ | ✓ | | 65.2 | 35.5 | 38.3 | 28.1 | 33.3 | 61.9 | 25.0 | 67.3 | 73.3 | 1820 |
| 4 | | ✓ | ✓ | ✓ | 65.0 | 35.5 | 38.7 | 28.4 | 33.3 | 62.0 | 25.5 | 67.6 | 73.6 | 1827 |
| 5 | ✓ | ✓ | ✓ | ✓ | 65.5 | 35.8 | 38.9 | 28.6 | 33.5 | 62.2 | 25.7 | 67.9 | 73.7 | 1836 |

our method continues to maintain a leading advantage across most datasets, demonstrating that the proposed DS-VLM exhibits excellent generalizability and possesses strong potential for adaptation to a wide range of VLM architectures.

## 4.3. Quantitative Comparison with SoTAs

We further compare our method with several leading approaches. The methods included in the comparison are MiniGPT4 (Zhu et al., 2023), Qwen-VL (Bai et al., 2023), VisualGLM (GLM et al., 2024), PandaGPT (Su et al., 2023), mPLUG-Owl2 (Ye et al., 2024), Emu2-chat (Sun et al., 2024), Yi-VL (Young et al., 2024) and ShareGPT-4V (Chen et al., 2025). Table 2 presents the performance comparison across multiple benchmarks.

Remarkably, despite relying solely on settings from LLaVA-1.5, our DS-VLM achieves performance that matches or surpasses the benchmarks set by leading SoTA methods, with a comparatively smaller volume of pretraining and instruction fine-tuning data.

## 4.4. Ablation Studies

**Supervised feature ablation** The ablation study of supervised features is shown in Table 3. The model selects Vicuna-13B as the Language Model and CLIP-L as the Vision Encoder. The experimental results demonstrate that combining text supervision (TS) and image supervision at different levels (IS) significantly improves the model's performance. Text supervision (TS) alone already leads to improvements, and progressively adding high-level, mid-level, and low-level image supervision further enhances the model's performance, particularly in terms of detail capture and global semantic understanding. The optimal combination is using text supervision along with multi-level image supervision (TS + IS High + IS Mid + IS Low), which performs best across multiple datasets, especially achieving the highest scores on metrics such as SQA and MME. This indicates that multi-level supervision from both text and image modalities can complement each other, comprehensively enhancing the model's multi-task learning ability and multimodal understanding capability.

**Reconstruction Loss Ablation** Additionally, to explore which reconstruction loss performs best in improving model performance, we conducted ablation experiments on reconstruction losses, with the results shown in Table 4. Specifically, the model uses Vicuna-7B as the LLM and CLIP-L as the visual encoder. We evaluated reconstruction losses at three levels of granularity: Mean Absolute Error (MAE), Structural Similarity Index (SSIM), and Perceptual Loss (PL). The experimental results across five datasets show that Perceptual Loss (PL) consistently achieved the best performance. This indicates that Perceptual Loss can help the supervised features retain more image details and semantic information, thereby enhancing the model's performance.

*Table 4.* Ablation study results on different reconstruction losses. MAE, SSIM and PL denote mean absolute error, structural similarity index and perceptual loss, respectively.

| RL | MMB | MMS | MMMU | MV | AI2D | SQA |
|---|---|---|---|---|---|---|
| MAE | 60.4 | 33.5 | 36.1 | 25.8 | 55.4 | 68.7 |
| SSIM | 61.1 | 34.3 | 36.6 | 26.2 | 56.0 | 69.3 |
| PL | 61.5 | 34.9 | 37.8 | 27.1 | 56.7 | 69.7 |

**Ablation Study on the Effectiveness of the Independent Cross-Attention Mechanism** To further assess the effectiveness of our proposed Independent Cross-Attention mechanism, we conduct an ablation study comparing it against the commonly used Shared Cross-Attention strategy. As illustrated in Table 5, the independent cross-attention design consistently outperforms the shared variant across all benchmark evaluations. These performance gains are primarily attributed to the modular separation of attention heads for textual and visual features, which facilitates more precise and independent modeling of modality-specific semantics. In contrast, the shared cross-attention approach is more susceptible to modality interference, thereby reducing the efficiency of feature fusion. The results demonstrate that the Independent Cross-Attention mechanism enables more robust multimodal alignment and enhances the model's reasoning and comprehension capabilities across a diverse set of tasks.

**Ablation Study on Adapter Architecture** To evaluate the effectiveness of our proposed linear adapter, we conduct an

*Table 5.* Ablation study results of different Cross-Attention mechanisms. SCA and ICA denote Shared Cross Attention and Independent Cross Attention, respectively.

| Method | MMB | MMS | MMMU | MV | AI2D | SQA |
|--------|------|------|------|------|------|------|
| SCA | 64.1 | 34.4 | 37.6 | 28.3 | 61.7 | 73.4 |
| ICA | 65.5 | 35.8 | 38.9 | 28.6 | 62.2 | 73.7 |

ablation study by replacing it with the Q-Former adapter utilized in BLIP-2 (Li et al., 2023b). As shown in Table 6, our linear adapter consistently outperforms the Q-Former-based design across six diverse vision-language benchmarks. Unlike Q-Former, which introduces additional query token mechanisms, the linear adapter employs a more parameter-efficient structure without increasing architectural complexity. These results validate the superiority of our lightweight and efficient adapter design.

*Table 6.* Ablation study results of different adapter structures.

| Method | MMB | MMS | MMMU | MV | AI2D | SQA |
|--------|------|------|------|------|------|------|
| Q-Former | 64.3 | 34.6 | 37.4 | 28.0 | 61.5 | 73.1 |
| MLP(Ours) | 65.5 | 35.8 | 38.9 | 28.6 | 62.2 | 73.7 |

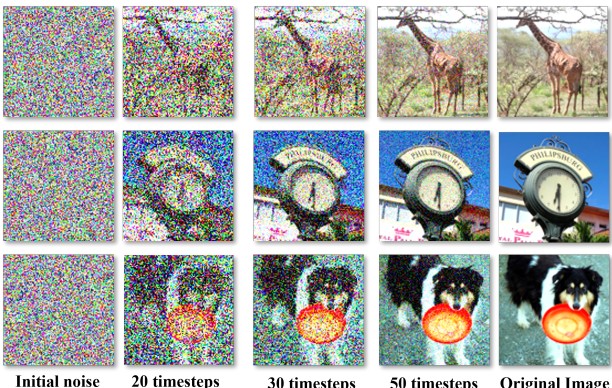

Initial noise   20 timesteps   30 timesteps   50 timesteps   Original Image

*Figure 5.* The figure illustrates the process by which the diffusion model gradually reconstructs a clear image from random noise over multiple timesteps.

### 4.5. Qualitative Analysis via Visualization

To verify the effectiveness of our proposed method, we conducted a visualization analysis of the image generation process of the diffusion model under given supervised features, as shown in Figure 5. Specifically, we present the image evolution at different stages of the denoising process to illustrate how the diffusion model progressively restores high-quality images. Starting from an initial random noise image, as the denoising timesteps progress, structural information gradually emerges, contours become clearer, details are continuously enriched, and eventually, a high-quality image with complete semantic information is generated. This

process fully demonstrates that supervised features play a crucial role in the generation process of the diffusion model. They provide rich semantic information, enabling the model to effectively recover image details and global structure during the denoising process. Experimental results further validate that the proposed DS-VLM framework can learn visual representations more efficiently through a shorter knowledge propagation chain, thereby achieving superior performance in image understanding tasks.

### 5. Conclusion

In this paper, we propose a novel Diffusion-Supervised Vision-Language Model (DS-VLM), which addresses the issue of information loss caused by the lengthy propagation path of text-based supervision signals in traditional methods by directly leveraging input images to supervise the visual encoder and connector through a diffusion model. Experimental results demonstrate that DS-VLM significantly improves the quality of feature extraction and achieves superior performance across various tasks and datasets. This approach provides an efficient and generalizable solution for multimodal learning, and its potential applications in broader scenarios can be further explored in the future.

### Acknowledgements

This work was supported by the National Science Fund for Distinguished Young Scholars (No.62025603), the National Natural Science Foundation of China (No. U21B2037, No. U22B2051, No. U23A20383, No. U21A20472, No. 62176222, No. 62176223, No. 62176226, No. 62072386, No. 62072387, No. 62072389, No. 62002305 and No. 62272401), and the Natural Science Foundation of Fujian Province of China (No. 2021J06003, No.2022J06001).

### Impact Statement

This paper presents work whose goal is to advance the field of Machine Learning. There are many potential societal consequences of our work, none which we feel must be specifically highlighted here.

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

## A. Comparative Experiments with LLaVA with Trainable Encoder

To further validate the effectiveness of our proposed DS-VLM under the setting of a trainable visual encoder, we conduct an additional set of comparative experiments to evaluate the overall performance of DS-VLM against the mainstream method LLaVA-1.5 when the visual encoder is trainable. Specifically, we perform comprehensive evaluations under two different visual encoder configurations—CLIP-L and SigLIP-SO—while keeping the language model (Vicuna-7B) and the training data (0.5M + 0.6M) consistent. As shown in Table 7, DS-VLM significantly outperforms LLaVA-1.5 across most evaluation benchmarks. This demonstrates that our proposed diffusion-based supervision mechanism can more effectively optimize the visual encoder through a shorter gradient propagation path, thereby improving the quality of multimodal semantic alignment and enhancing the model's overall reasoning capabilities.

*Table 7.* Comparative Study with LLaVA Framework Equipped with a Trainable Encoder

| Method | LM | VE | PT + IT | MMB | MMS | MMMU | MV | OCRB | AI2D | HB | LB | SQA | MME |
|---|---|---|---|---|---|---|---|---|---|---|---|---|---|
| LLaVA-1.5 | Vicuna-7B | CLIP-L | 0.5M + 0.6M | 59.4 | 33.6 | 36.2 | 26.3 | 31.5 | 55.8 | 27.8 | 62.2 | 68.9 | 1817 |
| DS-VLM | Vicuua-7B | CLIP-L | 0.5M + 0.6M | 61.5 | 34.9 | 37.8 | 27.1 | 32.4 | 56.7 | 28.9 | 64.3 | 69.7 | 1783 |
| LLaVA-1.5 | Vicuua-7B | SigLIP-SO | 0.5M + 0.6M | 62.6 | 35.2 | 38.1 | 27.2 | 36.6 | 58.7 | 27.5 | 67.0 | 70.4 | 1782 |
| DS-VLM | Vicuna-7B | SigLIP-SO | 0.5M + 0.6M | 63.2 | 36.8 | 38.4 | 27.6 | 37.5 | 60.0 | 27.8 | 67.7 | 71.1 | 1814 |

## B. Validating the Critical Role of the Vision Encoder in the Training Process

To verify that the optimization process in the DS-VLM framework is not solely dominated by the adapter module, we conducted a controlled experiment comparing two configurations: training only the adapter (with the vision encoder frozen) and jointly training both the vision encoder and the adapter. As shown in table 8, the reconstruction loss was 0.5 when only the adapter was trained, whereas it significantly decreased to 0.1 when both the vision encoder and the adapter were trainable. This result indicates that relying solely on the adapter is insufficient for effective visual reconstruction, and that the vision encoder indeed learns meaningful representations during training. This experiment effectively rules out the hypothesis that the adapter dominates the training process and further validates the effectiveness and necessity of our proposed diffusion supervision mechanism in optimizing the vision encoder.

*Table 8.* Reconstruction Loss under Different Freezing Strategies

| Method | Reconstruction Loss |
|---|---|
| Frozen Encoder+Trainable Adapter | 0.5 |
| Trainable Encoder+Trainable Adapter | 0.1 |

