# OpenReview forum: "DS-VLM: Diffusion Supervision Vision Language Model"
_ICML.cc/2025/Conference — ICML 2025 poster_

### Official Review · Reviewer_cFgh · 2025-02-23

**Overall Recommendation:** 4

**Summary:**

This paper proposed Diffusion Supervision Vision-Language Model (DS-VLM), which uses a diffusion module to provide additional supervision on the vision encoder and connector in the convential MLLM frameworks. Specifically, a frozen Stable Diffusion module takes the multi-level vision encoder features and the output of the connector as conditions to reconstruct the input image. It is claimed that DS-VLM facilitates the learning of visual semantic information of the vision encoder and connector. The experiments show that the newly introduced diffusion module can consistently improve the MLLM performance across different choices of vision encoders, LLM backbones, and training datasets.

**Claims And Evidence:**

The paper claims that the proposed DS-VLM can enrich the visual information learned by the vision encoder and connector. Their experiment results support this claim. The experiments are conducted across four different LLM backbones, two different vision encoders, and two different sets of training data. The method can improve the performance on most of the benchmarks under every setting.

**Essential References Not Discussed:**

N/A

**Experimental Designs Or Analyses:**

I checked the soundness/validity of the experimental designs and analysis. One issue is:
1. The original LLaVA framework freezes the vision encoder during training. However, this work fine-tunes the vision encoder. It is unclear whether the so-called "LLaVA" in Table 1 freezes the vision encoder or not. If yes, comparing the proposed method to LLaVA with a trainable vision encoder is necessary to verify the effectiveness.

**Methods And Evaluation Criteria:**

The proposed DS-VLM aims to address the long knowledge propagation chain in MLLM training. It uses a diffusion model to supervise the connector output and multi-level vision encoder features. Especially, the output of the connector is treated as text embedding and fed into the Text Adapter of the diffusion module. I agree that this supervision can help the connector learn more visual semantic information. However, the default input space of the Text Adapter (CLIP features) and the LLM backbone (word embeddings) are different. Why are the newly learned information assumed to be able to leveraged by the LLM backbone? As the experiment results suggest that the DS-VLM can indeed improve the performance, the discussion about this problem would bring valuable insights to the multimodal learning community.

**Other Comments Or Suggestions:**

Typos:
1. Additional slashes in Line 242-243.

Suggestions on writing:
1. Figure 3 shows that the LLM backbone is finetuned with LoRA. It would be better to also mention it in the Implementation Details in Section 4.1.
2. In Line 302-303, Section 4.1. The MLLM frameworks that use SigLIP as the vision encoder should be cited.
3. It would be better to also include the performance of LLaVA in Table 2.

**Other Strengths And Weaknesses:**

Other Strengths:
1. The proposed model outperforms the state-of the-art models in comparable sizes on a few benchmarks (Table 2).
2. The paper is well-written and easy to follow.

Other Weaknesses:
1. It would be appreciated if the paper can analyze qualitative results of the model and summarize on what types of questions the proposed method improves the performance. This might be helpful to support that the diffusion module enhances the learned semantic visual information.

**Questions For Authors:**

See **Methods And Evaluation Criteria** and **Experimental Designs Or Analysis**. Those are of my greatest concern.

**Relation To Broader Scientific Literature:**

Effective vision encoder and connector are essential to improve the MLLMs. While the previous works mainly focus on optimizing their architecture and training data, this work introduces additional training objectives on the vision encoder and connector. Moreover, the employment of the diffusion model also lies in the topic of representation learning via diffusion models. This paper extends the application of diffusion models to MLLM training.

**Theoretical Claims:**

This paper does not involve theoretical claims.

---

> ### Author Rebuttal · Authors · 2025-04-01
>
> > However, the default input space of the Text Adapter (CLIP features) and the LLM backbone (word embeddings) are different. Why are the newly learned information assumed to be able to leveraged by the LLM backbone?
> >
>
> A1: In response to the reviewer’s question, our DS-VLM framework leverages a diffusion model to supervise both the connector outputs and multi-level visual encoder features, effectively shortening the long knowledge propagation chain often seen in MLLM training.
>
> As for the difference between the Text Adapter’s input (CLIP features) and the LLM’s backbone (word embeddings), our key view is that the connector’s goal is to semantically align visual features to serve as effective textual embeddings. The diffusion model provides strong supervision and helps align feature distributions across modalities, allowing the connector to produce features that carry both visual detail and semantic meaning. These features, once projected into the LLM’s embedding space, can be naturally understood and used by the language model.
>
> Our experiments show that DS-VLM significantly improves multimodal task performance. We will further clarify this mechanism’s rationale and contributions in the final version.
>
> > The original LLaVA framework freezes the vision encoder during training. However, this work fine-tunes the vision encoder. It is unclear whether the so-called "LLaVA" in Table 1 freezes the vision encoder or not. If yes, comparing the proposed method to LLaVA with a trainable vision encoder is necessary to verify the effectiveness.
> >
>
> A2: Thank you for your suggestion. In Table 1, the LLaVA baseline freezes the visual encoder during training. Following your advice, we have conducted additional experiments comparing DS-VLM with a version of LLaVA where the visual encoder is trainable. As shown in the Table 1 below, DS-VLM still demonstrates clear advantages even when compared with LLaVA using a trainable visual encoder.
>
> Table1: Comparative experiments with LLaVA with trainable encoder
>
> | Method | LM | VE | PT+IT | MMB | MMS | MMMU | MV | OCRB | AI2D | HB | LB | SQA | MME |
> | --- | --- | --- | --- | --- | --- | --- | --- | --- | --- | --- | --- | --- | --- |
> | LLaVA-1.5 | Vicuua-7B | CLIP-L | 0.5M+0.6M | 59.4 | 33.6 | 36.2 | 26.3 | 31.5 | 55.8 | 27.8 | 62.2 | 68.9 | 1817 |
> | DS-VLM | Vicuua-7B | CLIP-L | 0.5M+0.6M | 61.5 | 34.9 | 37.8 | 27.1 | 32.4 | 56.7 | 28.9 | 64.3 | 69.7 | 1783 |
> | LLaVA-1.5 | Vicuua-7B | SigLIP-SO | 0.5M+0.6M | 62.6 | 35.2 | 38.1 | 27.2 | 36.6 | 58.7 | 27.5 | 67.0 | 70.4 | 1782 |
> | DS-VLM | Vicuua-7B | SigLIP-SO | 0.5M+0.6M | 63.2 | 36.8 | 38.4 | 27.6 | 37.5 | 60.0 | 27.8 | 67.7 | 71.1 | 1814 |
>
> > It would be appreciated if the paper can analyze qualitative results of the model and summarize on what types of questions the proposed method improves the performance. This might be helpful to support that the diffusion module enhances the learned semantic visual information.
> >
>
> A3: We agree that analyzing performance across question types helps validate the diffusion module’s impact. DS-VLM shows strong results on tasks requiring structural understanding and fine-grained perception, with notable gains on datasets like MMMU, MMB, and MMS. Specifically, MMMU covers multi-disciplinary visual reasoning (e.g., medicine and engineering), requiring accurate interpretation of complex visuals and domain-specific symbols—areas where DS-VLM excels due to its enhanced visual representations learned via pixel-level reconstruction. MMB evaluates broad multimodal capabilities, including structured diagrams and spatial layouts, where DS-VLM’s multi-layer supervision improves detail sensitivity. MMS further emphasizes semantic alignment between modalities, which benefits from our diffusion-enhanced visual features. We will include qualitative examples and analyses in the final version to better demonstrate DS-VLM’s effectiveness across these question types.
>
> > Additional slashes in Line 242-243.
> >
>
> A4: Thank you for pointing this out. We will remove the unnecessary slashes in lines 242–243 in the final version to avoid confusion.
>
> > Figure 3 shows that the LLM backbone is finetuned with LoRA. It would be better to also mention it in the Implementation Details in Section 4.1.
> >
>
> A5: Thank you for pointing out this issue. We will include the relevant information in the final version to ensure consistency between the text and figures, thereby improving the completeness and readability of the paper.
>
> > In Line 302-303, Section 4.1. The MLLM frameworks that use SigLIP as the vision encoder should be cited.
> >
>
> A6: We will include citations to MLLM frameworks that use SigLIP as the vision encoder in the final version to improve the discussion of related work.
>
> > It would be better to also include the performance of LLaVA in Table 2.
> >
>
> A7: Thank you for your suggestion. We will include the performance of LLaVA in Table 2 of the final version to enable a more comprehensive comparative analysis.

---

> > ### Comment · Reviewer_cFgh · 2025-04-04
> >
> > The authors' responses addressed my concerns. As a researcher working on training recipe of MLLMs, I highly understand that the feature alignment between visual embeddings and LLM input space, which is done by the connector, remains a challenging problem. Specifically, the basic mechanism of modern MLLMs is to connect pretrained LLMs with pretrained vision encoders through a lightweighted connector. The performance of MLLMs is notably affected by how well the connector can align two different modalities. In the conventional MLLM training recipe, the only supervision on this modality alignment is done through image captioning tasks or VQA tasks, which are high-level weak supervisions because not all the visual details are engaged in image captions and VQAs. The method proposed in this work makes a step forward in this challenge. Delivering the connector output to the text adaptor of a diffusion model serves as an auxiliary task to align visual space with text space. More importantly, by computing pixel-level reconstruction loss, this task provides low-level supervision on the connector. Although the diffusion model employed here does not have great technical novelty, I strongly appreciate the idea of leveraging pretrained diffusion models to address the challenging modality alignment problem in MLLMs.
> >
> > Therefore, I vote for the acceptance of this paper and raised my rating to 4 (accept).
> >
> > More suggestions to the authors: While Table 1 explores different visual encoders, they are both trained by vision-language tasks. In prior works, it has been found that visual encoders trained by self-supervised vision tasks (e.g., DINOv2) are more difficult to be aligned with the LLM backbone in MLLMs [1,2]. If DS-VLM can also facilitate the alignment of self-supervised visual encoders, it will make the work more impactful.
> >
> > [1] S. Tong et al. Cambrian-1. NeurIPS 2024.
> >
> > [2] S.  Karamcheti et al. Prismatic VLMs. ICML 2024.

---

> > > ### Author Response · Authors · 2025-04-07
> > >
> > > Thank you very much for your thoughtful comments and constructive suggestions. We truly appreciate your recognition of our work. Your proposed idea regarding the alignment of self-supervised visual encoders is highly insightful and valuable. We agree that verifying DS-VLM’s effectiveness in this setting would further enhance its significance. We will explore this direction and include relevant discussion in the final version to reflect its potential contributions to improving modality alignment in the MLLM community.

---

### Official Review · Reviewer_qD28 · 2025-03-13

**Overall Recommendation:** 3

**Summary:**

This paper proposes DS-VLM,  which introduces diffusion-based supervision to improve VLMs. To shorten the gradient propagation path, this paper reconstructs images using diffusion models, thereby improving supervision for the vision encoder and connector. Experimental results confirm the effectiveness of proposed method.

**Claims And Evidence:**

The paper reports good results across various test sets. Section 4.3 shows accuracy gains from different supervision features and Section 4.5 show typical reconstruction process for diffusion models. However, the connection between proposed method and performance improvements are still not clear, and more discussions are recommended.

**Essential References Not Discussed:**

NA

**Experimental Designs Or Analyses:**

This paper shows result improvement on multiple benchmarks. However, only the numbers are discussed in Section 4.2. More discussions on the improvement of model ability are recommended.

**Methods And Evaluation Criteria:**

The method and evaluation in this paper makes sense.

**Other Comments Or Suggestions:**

NA

**Other Strengths And Weaknesses:**

**Strength:**
1. This paper propose an interesting method that uses diffusion models as supervision to improve the VLM.
2. Experimental results show that the proposed method improves the performance over baseline model on multiple benchmarks.

**Weaknesses**:
1. The paper reports good results across various test sets. Section 4.3 shows accuracy gains from different supervision features and Section 4.5 show typical reconstruction process for diffusion models. However, the connection between proposed method and performance improvements are still not clear, and more discussions are recommended.

2. This paper shows result improvement on multiple benchmarks. However, only the numbers are discussed in Section 4.2. More discussions on the improvement of model ability are recommended.

3.  More ablation studies for different modules are recommended, such as the effect of adapters, the layer selection, cross-attentions.

4. Visualizations of model outputs are recommended to clearly show the difference between proposed method and baseline.

5. Line 324 mentions the Mini-Gemini dataset, where is the result?

6. Does the proposed method achives comparable results if trained on partial training data (such as 2/3)  of LLaVa?

**Questions For Authors:**

Please refer to weaknesses.

**Relation To Broader Scientific Literature:**

This paper improves the VLM by diffusion supervision, which may inspire the VLM community.

**Theoretical Claims:**

NA

---

> ### Author Rebuttal · Authors · 2025-04-01
>
> Thanks for your feedback. We’ll address each point in detail.
>
> > The paper reports good results across various test sets. Section 4.3 shows accuracy gains from different supervision features and Section 4.5 show typical reconstruction process for diffusion models. However, the connection between proposed method and performance improvements are still not clear, and more discussions are recommended.
> >
>
> > This paper shows result improvement on multiple benchmarks. However, only the numbers are discussed in Section 4.2. More discussions on the improvement of model ability are recommended.
> >
>
> A1: Thank you for pointing this out. The performance gains come from our diffusion-based pixel-level supervision, which creates a short gradient propagation chain from image pixels to visual features, greatly enhancing the visual encoder and connector’s representation capabilities.
>
> Our method incorporates multi-level visual feature supervision (high, middle, and low layers) along with an MOE-based fusion mechanism, allowing the model to comprehensively perceive semantic details and structural information within the image—effectively compensating for the semantic sparsity of traditional text-only supervision in VLMs. Furthermore, the visualization results in Section 4.5 clearly demonstrate how DS-VLM progressively reconstructs the original image, providing intuitive evidence of the effectiveness of supervised features in guiding the visual understanding process.
>
> The 1.9% gain on MMMU highlights our method’s strength in structured image understanding and semantic reasoning. We’ll clarify this causal link in the final version.
>
> > More ablation studies for different modules are recommended, such as the effect of adapters, the layer selection, cross-attentions.
> >
>
> A2: As shown in Table 3 (No.2–4) of the paper, we explored layer choices in our ablation study. Joint supervision across visual layers significantly boosts performance, as the layers complement each other and enhance supervision quality, improving model understanding and generalization.
>
> Following your suggestion, we conducted ablation studies on Adapter types (Table 1 below) and Cross Attention mechanisms (Table 2 below). Our proposed Adapter and decoupled Cross Attention show clear performance advantages. The multi-adapter structure enables finer adjustment of features from different modalities compared to the Q-Former-style Adapter. The decoupled Cross Attention avoids interference across feature levels, enabling each to fully express its semantic information through its dedicated attention path. These results validate the strength of our architectural design.
>
> Talbe1: Adapter type comparison
>
> | Method | MMB | MMS | MMMU | MV | OCRB | AI2D | HB | LB | SQA | MME |
> | --- | --- | --- | --- | --- | --- | --- | --- | --- | --- | --- |
> | Q-Former | 64.3 | 34.6 | 37.4 | 28.0 | 32.8 | 61.5 | 25.2 | 66.8 | 73.1 | 1805 |
> | Ours(MLP) | 65.5 | 35.8 | 38.9 | 28.6 | 33.5 | 62.2 | 25.7 | 67.9 | 73.7 | 1836 |
>
> Table2: Comparison using shared Cross-Attention mechanism
>
> | Method | MMB | MMS | MMMU | MV | OCRB | AI2D | HB | LB | SQA | MME |
> | --- | --- | --- | --- | --- | --- | --- | --- | --- | --- | --- |
> | Shared Cross attention | 64.1 | 34.4 | 37.6 | 28.3 | 33.1 | 61.7 | 24.9 | 67.0 | 73.4 | 1821 |
> | Ours(Decoupling Cross Attention) | 65.5 | 35.8 | 38.9 | 28.6 | 33.5 | 62.2 | 25.7 | 67.9 | 73.7 | 1836 |
>
> > Visualizations of model outputs are recommended to clearly show the difference between proposed method and baseline.
> >
>
> A3: We agree that visual comparisons are important for validating our method. In a fine-grained description task (e.g., “Describe the accessories worn”), the baseline missed the earring detail, while DS-VLM correctly generated “a silver hoop earring on the left ear.” This shows the benefit of our diffusion-based supervision in capturing fine details. We will add visual examples to highlight DS-VLM’s strength in fine-grained understanding.
>
> > Line 324 mentions the Mini-Gemini dataset, where is the result?
> >
>
> A4: The Mini-Gemini dataset is included in the "Expanding to larger training datasets" section of Table 1 in the paper. We'll clarify this in the final version.
>
> > Does the proposed method achives comparable results if trained on partial training data (such as 2/3) of LLaVa?
> >
>
> A5: As shown in Table 3 below, reducing training data leads to performance drop, highlighting the importance of full data for effective diffusion-based supervision. This analysis will be added in the final version.
>
> Table3: Comparison using partial training data
>
> | Method | LM | VE | PT+IT | MMB | MMS | MMMU | MV | OCRB | AI2D | HB | LB | SQA | MME |
> | --- | --- | --- | --- | --- | --- | --- | --- | --- | --- | --- | --- | --- | --- |
> | DS-VLM | Vicuna-7B | CLIP-L | 0.3M+0.4M | 60.7 | 34.8 | 37.3 | 26.4 | 32.3 | 56.5 | 28.7 | 63.9 | 70.3 | 1763 |
> | DS-VLM | Vicuna-7B | CLIP-L | 0.5M+0.6M | 61.3 | 35.5 | 38.1 | 26.7 | 32.6 | 56.9 | 29.2 | 65.0 | 70.6 | 1779 |

---

### Official Review · Reviewer_dHBy · 2025-03-14

**Overall Recommendation:** 2

**Summary:**

VLMs integrate visual and textual information to perform tasks such as image captioning, visual question answering, and image-text retrieval. However, current VLMs face two critical limitations: degraded supervision due to information loss during gradient propagation through LLMs, and the inherent semantic sparsity of textual supervision compared to visual data. To address these issues, the authors propose the Diffusion Supervision Vision-Language Model (DS-VLM), which uses a diffusion model to directly supervise the visual encoder and connector components, bypassing the LLM and providing richer semantic information from images.

**Claims And Evidence:**

The claims made in the submission are generally supported by empirical evidence. The authors provide experimental results, ablation studies, and visualizations to validate their proposed DS-VLM framework.

**Essential References Not Discussed:**

The paper provides a good review of related work. However, there are a few related works and concepts that are relevant but not currently cited or discussed in the paper:

1. Houlsby, N., Giurgiu, A., Jastrzebski, S., Morrone, B., Le Scao, K., Wolf, T., ... & Riedel, S. (2019). Parameter-efficient transfer learning for NLP. International Conference on Machine Learning, 2790-2799.
This work provides a theoretical and practical foundation for using adapter modules to enhance model performance without extensive retraining.

2. Li, J., Chen, D., Hoi, S., & Lu, J. (2019). VisualBERT: A simple and strong visual-language fusion model. Proceedings of the 57th Annual Meeting of the Association for Computational Linguistics, 1-12.
3. Lu, J., Batra, D., Parikh, D., & Lee, S. (2019). ViLBERT: Pretraining task-agnostic visual-language representations for vision-language tasks. Advances in Neural Information Processing Systems, 32, 13-23.
These models laid the groundwork for integrating visual and textual information using transformer architectures, which is relevant to the design of DS-VLM.

**Experimental Designs Or Analyses:**

The experimental designs and analyses are generally sound. The authors have conducted a set of experiments to validate their proposed DS-VLM framework, including comparisons with baselines, ablation studies, and visualizations. The results demonstrate improvements over baseline models and state-of-the-art methods, supporting the claims made. The suggestions for additional analyses (e.g., statistical significance, computational overhead) would further strengthen the paper and provide a better evaluation of the proposed method.

**Methods And Evaluation Criteria:**

The proposed methods and evaluation criteria are suited for the problem of improving vision-language alignment in VLMs. The DS-VLM framework is technically sound, addressing the limitations of traditional supervision methods. The evaluation criteria, including benchmark datasets, performance metrics, ablation studies, and visualizations, provide a assessment of the method's effectiveness. The results demonstrate improvements over baseline models and state-of-the-art methods, validating the claims made by the authors.

**Other Comments Or Suggestions:**

N/A

**Other Strengths And Weaknesses:**

Pro：
1. This paper is generally well-written.
2. The experiments demonstrate the effectiveness of their proposed method.
3. The authors also conduct a set of ablation studies to better understand the proposed method.

Con:
1. While the idea of using diffusion models to supervise vision-language models is interesting, it appears to be an incremental extension of existing work rather than a new approach. For instance, the integration of diffusion models in VLM has been extensive explored in previous studies such as DiffTPP, and also been explored in other contexts (e.g., text-to-image generation), and the application to VLMs seems like a straightforward adaptation.
2. The core components of DS-VLM, such as the Multi-Adapter Diffusion model and the MOE Cross Attention mechanism, are built on well-established concepts. The paper does not introduce any new mechanisms or architectures that advance the field.
3. The training process involving diffusion models is computationally intensive. The paper does not provide any analysis on the training time and resource requirements, which are critical for practical applications. This lack of information raises concerns about the feasibility of deploying DS-VLM in real-world scenarios.

**Questions For Authors:**

See "Other Strengths And Weaknesses"

**Relation To Broader Scientific Literature:**

The key contributions of the DS-VLM paper are aligned with the broader scientific literature on VLM, diffusion models, and multimodal learning. The paper leverages recent advancements in diffusion models to address a few limitations in traditional VLMs, providing a solution for enhancing visual feature extraction. By building on prior work, DS-VLM represents one step forward in the development of robust and efficient multimodal models.

**Theoretical Claims:**

N/A

---

> ### Author Rebuttal · Authors · 2025-04-01
>
> > The suggestions for additional analyses (e.g., statistical significance, computational overhead) would further strengthen the paper and provide a better evaluation of the proposed method.
> >
>
> > The training process involving diffusion models is computationally intensive. The paper does not provide any analysis on the training time and resource requirements, which are critical for practical applications. This lack of information raises concerns about the feasibility of deploying DS-VLM in real-world scenarios.
> >
>
> A1: Thank you for your constructive suggestion. We conducted full training of DS-VLM on 8 H100 GPUs. As shown in Table 1 below, compared to the original LLaVA-1.5, DS-VLM increases training time by approximately 15%. However, during inference, the diffusion module is not involved, so there is no additional inference overhead compared to the baseline. Given the performance gains on multiple benchmarks (e.g., a 2.1% improvement on MMMU and 1.8% on MMS), we believe this moderate training cost is justified. We will include this analysis in the final version to better illustrate the practicality of our method.
>
> Talbe1: Comparison of two-Stage training time consumption under different settings.
>
> |  | Vicuna-7B |  | Vicuna-13B |  |
> | --- | --- | --- | --- | --- |
> |  | PT | IT | PT | IT |
> | LLaVA-1.5 | 2.0 | 6.7 | 2.8 | 10.8 |
> | DS-VLM | 2.5 | 7.8 | 3.4 | 12.1 |
>
> > The paper provides a good review of related work. However, there are a few related works and concepts that are relevant but not currently cited or discussed in the paper:
> >
>
> A2: Thank you for highlighting the missing references. We appreciate your suggestions and will include and discuss them in the final version.
>
> > While the idea of using diffusion models to supervise vision-language models is interesting, it appears to be an incremental extension of existing work rather than a new approach. For instance, the integration of diffusion models in VLM has been extensive explored in previous studies such as DiffTPP, and also been explored in other contexts (e.g., text-to-image generation), and the application to VLMs seems like a straightforward adaptation.
> >
>
> A3: Thank you for your valuable comments. We understand the concern about the novelty of using diffusion to guide VLMs. Our approach is not a direct application of existing methods but an innovative integration of diffusion into VLMs.
>
> Unlike methods such as DiffTPP, which uses diffusion for generation, we first use it as a supervision signal during training. This enables the construction of a short-path gradient propagation chain from the image pixel space to visual features. Such a mechanism effectively mitigates the supervision information loss caused by long gradient paths in current VLM training. At the same time, the image reconstruction task provides fine-grained, multi-level optimization signals for both the visual encoder and the connector.
>
> Furthermore, our proposed Multi-Adapter Diffusion architecture flexibly receives supervision features from different modalities and employs MOE Cross Attention to aggregate information across multiple layers—a design that, to the best of our knowledge, is unprecedented in prior work.
>
> Therefore, we believe this work introduces substantial and original contributions in coupling diffusion models with VLMs, rather than being a generic application of diffusion techniques to VLMs. We will clarify this point further in the final version of the paper.
>
> > The core components of DS-VLM, such as the Multi-Adapter Diffusion model and the MOE Cross Attention mechanism, are built on well-established concepts. The paper does not introduce any new mechanisms or architectures that advance the field.
> >
>
> A4: Thank you for your thoughtful review. We sincerely acknowledge that while the Multi-Adapter Diffusion and MOE Cross Attention mechanisms in DS-VLM are inspired by some existing foundational concepts, our method introduces key innovations in how these components are integrated and applied. Specifically, Multi-Adapter Diffusion is the first to leverage a diffusion model for supervising intermediate features during the training phase. It receives features from multiple layers of the visual encoder as well as from the connector, and reconstructs the image through a unified U-Net framework. Meanwhile, the MOE Cross Attention mechanism processes these heterogeneous features through separate attention pathways and dynamically fuses them via expert routing, enabling the injection of fine-grained supervision signals. This design not only establishes a short supervision path from image pixels to multi-level visual features but also significantly enhances the structural coherence and semantic expressiveness of the learned representations. We believe this constitutes a substantial advancement over traditional VLM optimization paradigms. We will further highlight the uniqueness and novelty of these mechanisms in the final version of the paper.

---

> > ### Comment · Reviewer_dHBy · 2025-04-02
> >
> > I sincerely appreciate the authors' responses. Although the authors have provided explanations, in terms of methodology, this paper is built upon existing concepts or techniques. While there are differences in implementation, mere implementation differences are not sufficient for acceptance at a top-tier conference like ICML. Therefore, I will maintain my original score.

---

> > > ### Author Response · Authors · 2025-04-07
> > >
> > > Thank you for your feedback. We agree that our method is partially built upon existing techniques with corresponding improvements. However, our work introduces a key improvement over conventional high-level weak supervisions (e.g., captioning, VQA) by incorporating low-level pixel-wise supervision through diffusion-based reconstruction. This provides a denser and more precise learning signal for modality alignment between vision and language. We believe this direction offers valuable insights for the MLLM community and opens up new possibilities for developing more robust and generalizable multimodal models. We will give more discussion in the revised version. Thank you again for your valuable time and thoughtful review.

---

### Official Review · Reviewer_gwC3 · 2025-03-14

**Overall Recommendation:** 3

**Summary:**

The authors design a framework to train the visual encoders in VLMs, with diffusion supervision. The authors design a mechanism to incorporate visual features from multiple levels of the encoder to reconstruct the image using a frozen diffusion model during training.  The authors evaluate their method, built on top of existing VLMS, on various VLM benchmarks and demonstrate some improvement over the VLM by itself.

**Claims And Evidence:**

The improvements are marginal ~1-2% and not "substantial" as claimed by the authors. Also for many benchmarks there is no improvement at all in Table 1.

**Essential References Not Discussed:**

NA

**Experimental Designs Or Analyses:**

In Table 2 the authors show the best results using the proposed framework + Qwen2-7B but they do not show results on Qwen2-7B by itself. Also it is not the best across all benchmarks. The authors need to analyse this inconsistency in performance a cross the different benchmarks.

**Methods And Evaluation Criteria:**

It is not clear to me how the authors train their framework. Typically diffusion training involves sampling a noise level and predicting and supervising the noise. Instead the authors say that they supervise the image with reconstruction loss, which would mean the authors need to make several iterative passes through the Diffusion model. Do the authors roll out the diffusion process during training to backpropagate gradients. This needs to be clarified in the paper.

The authors add a bunch of new layers to feed the multi-level features into the diffusion model. I assume these layers are not used during inference since their only purpose is to adapt the features for diffusion model. Seems to me that this could result in the  primary encoder not learning anything useful since the new layers(unused during inference) maybe doing most of the heavy lifting. Have the authors figured out a way to solve this issue ?

**Other Comments Or Suggestions:**

line 242-243, "/////" added for no reason.

**Other Strengths And Weaknesses:**

NA

**Questions For Authors:**

See above.

**Relation To Broader Scientific Literature:**

The authors introduce a new diffusion based framework to finetune the Vision Encoder in VLMs. This method is not particularly new except the proposed use of Diffusion models, as illustrated by the authors in Fig 2.

**Theoretical Claims:**

NA

---

> ### Author Rebuttal · Authors · 2025-04-01
>
> > The improvements are marginal ~1-2% and not "substantial" as claimed by the authors. Also for many benchmarks there is no improvement at all in Table 1.
> >
>
> > The authors need to analyse this inconsistency in performance a cross the different benchmarks.
> >
>
> A1: Thank you for pointing out the modest improvements and performance inconsistency. We understand your concern and clarify that DS-VLM is not just about fine-tuning gains, but proposes a new optimization paradigm: diffusion-based image reconstruction supervision. It delivers dense semantic signals from pixel space to the encoder and connector, shortening the gradient path in traditional text supervision and mitigating semantic sparsity, thus enhancing structural and semantic representation. This is particularly effective on datasets like MMMU, MMBench, and MMStar. In OCRB tasks requiring high-res text recognition, our reconstruction focuses on structure and semantics, lacking detail for small text. We plan to explore enhanced text-area reconstruction in future work. DS-VLM adds no extra inference cost, ensuring practicality and scalability. We will include this analysis in the final version.
>
> > Do the authors roll out the diffusion process during training to backpropagate gradients. This needs to be clarified in the paper.
> >
>
> A2: The gradient backpropagation path in DS-VLM follows: Image Reconstruction Loss → Diffusion → Adapter → Vision Encoder. Below, we derive the backpropagation process of the model. For simplicity, we set the number of diffusion iterations to 2. We begin by defining:
>
> $z_e=E(f;a)$
>
> $z_a=A(z_e;b)=A(E(f;a);b)$
>
> $z_{d1}=D(x,z_a;c)=D(x,A(E(f;a);b);c)$
>
> $z_{d2}=D(z_{d1},z_a;c)=D(D(x,A(E(f;a);b);c),A(E(f;a);b);c)$
>
> $L=L(z_{d2})$
>
> Here, $E$ denotes the encoder with parameters $a$, $A$ denotes the adapter with parameters $b$, and $D$ denotes the diffusion module with parameters $c$. $L$ denotes the reconstruction loss.
>
> According to the chain rule, the gradient of the Adapter parameter can be deduced as:
>
> $\frac{\partial L}{\partial b}=\frac{\partial L}{\partial z_{d2}}[\frac{\partial D(z_{d1},z_a;c)}{\partial z_{d1}}\cdot \frac{\partial D(x,z_a)}{\partial z_a}+\frac{\partial D(z_{d1},z_a)}{\partial z_a}]\cdot \frac{\partial A}{\partial b}$
>
> The gradient of the Encoder parameters is：
>
> $\frac{\partial L}{\partial b}=\frac{\partial L}{\partial z_{d2}}[\frac{\partial D(z_{d1},z_a)}{\partial z_{d1}}\frac{\partial D(x,z_a)}{\partial z_a}+\frac{\partial D(z_{d1},z_a)}{\partial z_a}]\frac{\partial A(z_e;b)}{\partial z_e}\frac{\partial E(f;a)}{\partial a}$
>
> I hope the above can answer your questions. We look forward to further communication.
>
> > The authors add a bunch of new layers to feed the multi-level features into the diffusion model. I assume these layers are not used during inference since their only purpose is to adapt the features for diffusion model. Seems to me that this could result in the primary encoder not learning anything useful since the new layers(unused during inference) maybe doing most of the heavy lifting. Have the authors figured out a way to solve this issue ?
> >
>
> A3: To address concerns about the Adapter dominating learning, we ran a controlled experiment (Table 1 below) comparing: (1) freezing the encoder and (2) training both. Training both gave lower reconstruction loss, confirming the encoder is effectively optimized. The lightweight Adapter acts as a bridge to the diffusion model, supporting—not replacing—the encoder, ensuring efficient training and better visual representation.
>
> Table1
>
> | Method | Reconstruction Loss |
> | --- | --- |
> | Frozen encoder+Trainable adapter | 0.5 |
> | Trainable encoder+Trainable adapter | 0.1 |
>
> > In Table 2 the authors show the best results using the proposed framework + Qwen2-7B but they do not show results on Qwen2-7B by itself
> >
>
> A4: The experimental results for Qwen2-7B are presented in the third row of the "Expanding to other LLMs" section in Table 1 in the article. We will add these results to Table 2 in the final version.
>
> > This method is not particularly new except the proposed use of Diffusion models, as illustrated by the authors in Fig 2.
> >
>
> A5: Thank you for your interest. Instead of simply using diffusion for supervision, we propose a novel design to tackle long supervision paths and sparse textual signals in VLM training. Our Multi-Adapter Diffusion fuses multi-level visual and connector features, while MOE Cross Attention dynamically routes multimodal inputs for fine-grained fusion. Together, they create a compact, semantically rich supervision path that enhances visual representation. We will further clarify these contributions in the final version.
>
> > line 242-243, "/////" added for no reason.
> >
>
> A6: Thanks for pointing this out. The slashes will be removed in the final version.

---

> > ### Comment · Reviewer_gwC3 · 2025-04-04
> >
> > Based on the authors' response i have improved my score to weak-accept.

---

> > > ### Author Response · Authors · 2025-04-07
> > >
> > > We thank you for your final recognition of our work, and sincerely appreciate your thoughtful comments and support. The discussion in the rebuttal will be updated in the revised version.

---

### Decision · Program_Chairs · 2025-05-01

**Decision:**

Accept (poster)

**Comment:**

This paper received mixed reviews: one Accept, two Weak Accepts, and one Weak Reject. Reviewer dHBy raised concerns that the work is primarily built upon existing concepts and techniques. In contrast, the Area Chair emphasized that a strong submission should address a practically meaningful and technically challenging problem. From this perspective, the issue of feature alignment between visual embeddings and the input space of large language models (LLMs)—as tackled by the proposed connector—remains a critical challenge in the development of multimodal large language models (MLLMs). Although the diffusion model utilized in this work may not introduce substantial technical novelty, the AC highly valued the insight of leveraging pretrained diffusion models to address the challenging problem of modality alignment in MLLMs. As such, the AC recommends accepting this paper.